# Biological Control Activities of Rhizosphere Fungus *Trichoderma virens* T1-02 in Suppressing Flower Blight of Flamingo Flower (*Anthurium andraeanum* Lind.)

**DOI:** 10.3390/jof10010066

**Published:** 2024-01-15

**Authors:** Dusit Athinuwat, On-Uma Ruangwong, Dulanjalee L. Harishchandra, Kitsada Pitija, Anurag Sunpapao

**Affiliations:** 1Department of Agricultural Technology, Faculty of Science and Technology, Thammasat University, Khlong Luang District, Pathum Thani 12120, Thailand; athinova6@hotmail.com; 2Department of Entomology and Plant Pathology, Faculty of Agriculture, Chiang Mai University, Mueang, Chiang Mai 50200, Thailand; on-uma.r@cmu.ac.th (O.-U.R.); dulanjalee.h@cmu.ac.th (D.L.H.); 3Office of Research Administration, Chiang Mai University, Chiang Mai 50200, Thailand; 4PerkinElmer Scientific (Thailand) Co., Ltd., 290 Soi Soonvijai 4, Bang Kapi, Huai Kwang, Bangkok 10310, Thailand; kitsada.pitija@perkinelmer.com; 5Agricultural Innovation and Management Division (Pest Management), Faculty of Natural Resources, Prince of Songkla University, Hatyai 90110, Thailand

**Keywords:** antibiosis, biocontrol agent, competition, parasitism, VOCs

## Abstract

Flower blight caused by *Neopestalotiopsis clavispora* is an emerging disease of flamingo flower (*Anthurium andraeanum* Lind.) that negatively impacts flower production. The use of rhizosphere fungi as biocontrol agents is an alternative way to control this disease instead of using synthetic fungicides. This research aimed to screen the potential of rhizosphere fungi, *Trichoderma* spp., with diverse antifungal abilities to control *N. clavispora* and to reduce flower blight in flamingo flowers. A total of ten isolates were tested against *N. clavispora* by dual culture assay, and T1-02 was found to be the most effective isolate against *N. clavispora*, with inhibition of 78.21%. Morphology and molecular phylogeny of multiple DNA sequences of the genes, the internal transcribed spacer (ITS), translation elongation factor 1-α (*tef1-α*), and RNA polymerase 2 (*rpb2*) identified isolate T1-02 as *Trichoderma virens*. Sealed plate method revealed *T. virens* T1-02 produced volatile antifungal compounds (VOCs) against *N. clavispora*, with inhibition of 51.28%. Solid-phase microextraction (SPME) was applied to trap volatiles, and GC/MS profiling showed VOCs emitted from *T. virens* T1-02 contained a sesquiterpene antifungal compound—germacrene D. The pre-colonized plate method showed that *T. virens* T1-02 aggressively colonized in tested plates with inhibition of 100% against *N. clavispora*, and microscopy revealed direct parasitism onto fungal hyphae. Furthermore, the application of *T. virens* T1-02 spore suspension reduced the disease severity index (DSI) of flower blight in flamingo flowers. Based on the results from this study, *T. virens* T1-02 displays multiple antagonistic mechanisms and has the potential ability to control flower blight of flamingo flowers caused by *N. clavispora*.

## 1. Introduction

The rhizosphere is the zone or region of soil surrounding the plant roots, and it is directly influenced by root activities and the uptake of nutrients and water [1]. The interaction between rhizosphere and plant roots plays an important role in soil health in natural and agricultural ecosystems [2,3]. The activity of soil microorganisms is generally enhanced by root exudates in the plant–rhizosphere system [4]. As soil fungal communities play a crucial role in plant growth and plant disease control [4], rhizosphere fungi are closely linked to plant growth and health by acting as antagonists against plant pathogens and plant growth promoters [5].

Filamentous fungi of the genus *Trichoderma* are known as common rhizosphere inhabitants [6]. *Trichoderma* species can produce a mass of conidia for asexual reproductive propagules [7]. *Trichoderma* species commonly inhabit surface soil, rhizosphere soil, and endophytes in plant tissues [8]. They have been widely used to control plant diseases due to their capacity to compete for nutrients and space [9], produce antimicrobial metabolites [10], possess valuable volatile compounds [11,12,13] and mycoparasites against plant pathogens [14,15], induce plant resistance against disease [16], and promote plant growth [17]. Due to the multifaceted mechanisms of *Trichoderma* spp., they have been used as biological control agents (BCAs) against several tropical plant diseases [17,18], such as stem canker of dragon fruit [15], gummy stem blight of muskmelon [16] and *Cercospora* leaf spot of sugar beet [18]. Hence, these rhizosphere fungi are a great source of natural fungicides, due to their potential to produce beneficial compounds or metabolites from those rhizosphere fungi that can be used to protect their plant host from plant pathogens.

Flamingo flower (*Anthurium andraeanum* Lind.) is a flowering plant belonging to the family Araceae, a monocotyledonous perennial that prefers warm humid climates in the tropical and subtropical regions. This plant is normally grown under humid and moderate temperature conditions, which are commonly found in Thailand, especially in the southern region. However, due to the tropical climate, flamingo flowers face several plant diseases caused by fungal phytopathogens and fungal-like organisms, for instance, black nose caused by *Colletotrichum theobromicola* [19], leaf spot caused by *Myrothecium roridum* [20], and stem rot caused by *Phytopythium vixans* [21].

Recently, an emerging disease of flamingo flowers has been noted in southern Thailand, including spadix rot caused by *Lasiodiplodia theobromae* [22] and flower blight caused by *Neopestalotiopsis clavispora* [23]. Flower blight is one of the emerging diseases that causes great loss in both the quality and quantity of flower production in southern Thailand [23]. The disease spread is about 20% in plantation areas and negatively impacts market value. As accurate identification of causal agents is considered the first step for disease management, finding appropriate methods to control the disease is an important next step.

Although the use of synthetic fungicides effectively reduces loss from diseases [24], extensive use of chemical fungicides leads to resistant fungal pathogens [25], can cause residue accumulation in the ecosystems [26], and causes harmful side effects on human health [27]. Developing biological control and management methods can be an effective strategy to control flower blight and may benefit disease management in flamingo flowers. The study of interactions between plant hosts and rhizosphere fungi provides basic knowledge about managing plant pests. However, biocontrol mechanisms of new strains of rhizosphere fungi of the genus *Trichoderma* are still of high interest in agricultural research, because they carry some valuable metabolites that can be used to explore disease control strategies in the plant pathology field. Therefore, the aim of this research was to screen and test the fungicidal activities of rhizosphere *Trichoderma* species against *N. clavispora*, the pathogen of flower blight of flamingo flowers.

## 2. Materials and Methods

### 2.1. Fungal Strains

A total of ten *Trichoderma* strains—K1-01, K2-02, K3-01, T1-02, T3-01, Z1-01, Z1-02, Z2-02, SP1-01, and SP2-02—had been isolated from rhizosphere soil of Para rubber tree (*Hevea brasiliensis*) in a plant-protected area in Songkhla province, southern Thailand using a soil dilution pour plate in a previous study [6] and deposited in the Culture Collection of the Pest Management Department, Faculty of Natural Resources, Prince of Songkla University, Thailand. The flower blight pathogen *N. clavispora* strain PSU-AAN2 was isolated from the flamingo flower in Thailand [23]. Both *Trichoderma* strains and *N. clavispora* strain PSU-AAN2 were obtained from the Culture Collection of the Pest Management Department, Faculty of Natural Resources, Prince of Songkla University, Thailand. *Trichoderma* spp. and *N. clavispora* were cultured on potato dextrose agar (PDA) and incubated at 28 ± 2 °C for 3 days before being used in this study.

### 2.2. Dual Culture Assay

An agar plug of a 3-day-old *N. clavispora* colony was placed on the side of a 9 cm petri dish, with an agar plug of each strain of *Trichoderma* spp. placed on the opposite side 5 cm away from the pathogen. PDA plates with pathogen alone served as the control. The experiment was designed as a complete randomized design (CRD) with five replicates and was repeated twice. The tested plates were incubated at ambient temperature (28 ± 2 °C) for 5 days. Colony radii of *N. clavispora* were measured and the percentage inhibition was calculated with the formula:$$\mathrm{Percentage}\;\mathrm{of}\;\mathrm{inhibition}\;=\frac{R1-R2}{R1}\times 100$$
where R1 is the colony radii of *N. clavispora* in the control plate and R2 is the colony radii of *N. clavispora* in tested plates [28].

### 2.3. Identification of Trichoderma Species

Selected *Trichoderma* isolates with the highest percentage inhibition against *N. clavispora* were cultured on PDA and incubated at ambient temperature for 5 days for morphological study. Macroscopic and microscopic features of *Trichoderma* strains were observed using a stereomicroscope (Leica S8AP0, Leica Microsystems, Wetzlar, Germany) and compound microscope (Leica DM750, Leica Microsystems). Colony growth on PDA and morphological characteristics and dimensions of phialides and conidia were observed and measured (n = 20).

For molecular study, selected *Trichoderma* strains were cultured at ambient temperature for 2 days, and young mycelia were harvested and subjected to DNA extraction by mini-preparation [29]. Internal transcribed spacer (ITS), translation elongation factor 1-α (*tef1-α*), and RNA polymerase II (*rpb2*) sequences were amplified by PCR using a BIORAD T100TM Thermal Cycler (Bio-rad, Hercules, CA, USA). Sequences of ITS, *tef1-α* and *rpb2* were amplified by using primer pairs ITS1/ITS4 [30], EF1-728F/EF2 [31,32] and fRPB2-5F/fRPB2-7cr [33], respectively. The PCR reaction mixture contained 2× PCR DreamTaq Green PCR master mix (Thermo Scientific, Waltham, MA, USA), 10 pmol of each primer, DNA template, and nuclease distilled water (DW) in a 50 μL tube. The amplification temperature for initial denaturation was 94 °C for 3 min, followed by 35 cycles of denaturation at 94 °C for 1 min, annealing at 60 °C for 1 min, and extension at 72 °C for 3 min, with a final extension at 72 °C for 10 min. PCR products were visualized through 1% agarose gel electrophoresis. PCR products were sequenced at Macrogen Sequencing Service (Macrogen, Republic of Korea) with the same primers used in PCR amplification.

The chromatograms of each sequence were checked to verify their quality using BioEdit software v7.0.5.2 [34], and the ambiguous bases at the 5′ and 3′ ends were edited or deleted accordingly. Based on the results of BLASTn searches of GenBank with each of the newly sequenced isolates, sequences of species of *Trichoderma* were selected to be included in the phylogenetic analyses [35]. Individual datasets for each gene consisting of reference sequences and sequences obtained in the current study were aligned using the default settings of MAFFT 7 [36] and improved manually where necessary with BioEdit. The gene regions were combined in ITS, *rpb2* and *tef1*.

Phylogenetic analyses of the combined data set of *Trichoderma* spp. were performed by maximum likelihood (ML) and Bayesian inference analysis (BI). ML analysis was performed in RAxML [37] implemented in the CIPRES Science Gateway web server RAxML-HPC2on XSEDE [38] with 1000 rapid bootstrap replicates, using the GTRGAMMA model of nucleotide evolution. Maximum-likelihood bootstrap values (MLBS) equal to or greater than 50% are given above each node in the phylogenetic tree.

The best-fit nucleotide substitution model for BI for each gene was separately determined using MrModeltest v. 2.3 [39] under the Akaike information criterion (AIC) implemented in PAUP v. 4.0b10. BI analysis was performed using the Markov Chain Monte Carlo (MCMC) method with MrBayes 3.2.2 [40]. Five million generations were run, with sampling frequency at every 1000th generation. Twenty-five percent of the trees were discarded, representing the burn-in phase.

### 2.4. Volatile Antifungal Bioassay

Several *Trichoderma* strains are able to produce and release volatile organic compounds responsible for suppressing fungal growth. In order to test the effect of volatiles emitted from *Trichoderma* in restricting the growth of *N. clavispora* in PDA plates, the sealed plate method was used according to the methodology provided in Dennis and Webster [41]. The selected *Trichoderma* isolate was cultured on PDA in 9 cm petri dishes for 5 days. A mycelial plug was cut from cultured plates and placed centrally, and the lid of each petri dish was removed. The bottom plate was replaced with a plate inoculated with *N. clavispora*, and the two bottom plates were sealed together with parafilm. The bottom plate with PDA alone served as the control. The experiment was designed as CRD with five replicates and was repeated twice. Tested plates were incubated at ambient temperature (28 ± 2 °C) for 5 days. Colony diameters of *N. clavispora* were measured and converted to percentage inhibition using the following formula:$$\mathrm{Percentage}\;\mathrm{of}\;\mathrm{inhibition}\;=\frac{\mathrm{Dc}-\mathrm{Dt}}{\mathrm{Dc}}\times 100$$
where Dc is the mycelial growth of *N. clavispora* on the control plate and Dt is the mycelial growth of *N. clavispora* on the tested plate.

### 2.5. Solid-Phase Microextraction (SPME) Gas Chromatography–Mass Spectrometry (GC/MS) Analysis

To analyze volatile organic compounds (VOCs) emitted by *Trichoderma* spp., solid-phase microextraction (SPME) gas chromatography–mass spectrometry (SPME-GC/MS) was conducted [42]. The most effective strain of *Trichoderma* was cultured in a 20 mL chromatography vial (PerkinElmer, Waltham, MA, USA) and incubated at ambient temperature (28 ± 2 °C) for 7 days. Volatiles were extracted by SPME with headspace (HS) derivatization, and the extraction temperature was set to 125 °C for 25 min, needle temperature 130 °C for 1.5 min, and the transfer line 135 °C for 0.03 min. Separation of volatile compounds was conducted using a Clarus model 690 gas chromatograph (PerkinElmer, Waltham, MA, USA) coupled with an SQ8 mass-selective detector. The capillary column contained 5% phenylmethylpolysiloxane, 30 m × 250 μm ID × 0.25 μm film thickness (PerkinElmer, Waltham, MA, USA). The oven temperature was held at 60 ℃, then increased at a rate of 7 °C/min to a final temperature of 200 °C. The injector temperature was set at 200 °C and purified helium gas was used as the carrier gas at a flow rate of 1 mL/min. Electron impact (EI) mass spectra were collected at 70 eV ionization voltages over the range of *m*/*z* 45–550 and the electron multiplier voltage was set at 1400 V. The ion source and quadrupole temperatures were set at 200 °C. Volatile compounds were identified through a computer search of the National Institute of Standard and Technology (NIST) Mass Spectral Library Search Chromatogram.

### 2.6. Pre-Colonized Plate Method

To investigate the mycoparasitism of *Trichoderma* species, the pre-colonized plate method was used [10,15]. A mycelial plug of a 5-day-old colony of *N. clavispora* was cut and placed on one side of PDA and incubated at ambient temperature for 7 days. A strip of the most effective strain of *Trichoderma* (4 × 0.5 cm) was cut from a 7-day-old colony cultured on PDA, placed on the other side of PDA, and wholly pre-colonized by the isolate from *N. clavispora* plate. The tested plates were incubated at ambient temperature in dark conditions for 4 weeks. Ten agar plugs from the tested plates were then placed on 20% PDA to observe the growth of *Trichoderma* and *N. clavispora*. The percentage of colonization of each fungus was calculated. Growth of the *Trichoderma* colony in 20% PDA plates in comparison with the *N. clavispora* colony was counted and converted to percentage colonization. Mycoparasitism of *Trichoderma* was checked after 5 days of incubation with a compound microscope (Leica DM750, Leica Microsystems). After 10 days of incubation, parasitism of *Trichoderma* was checked through a thin polymer membrane nano-suit method [43], and was observed directly with a JSM-6100 SEM (JEOL, Peabody, MA, USA).

### 2.7. In Vivo Test

To test the efficacy of *Trichoderma* in reducing the flower blight of flamingo flower, an in vivo test was conducted with the flamingo flower Cheer cultivar. The selected *Trichoderma* isolates and *N. clavispora* were cultured on PDA for 7 days and incubated at ambient temperature (28 ± 2 °C) for sporulation. Conidia of both *Trichoderma* and *N. clavispora* were harvested. The conidial suspension of each fungus was adjusted in concentration as 10^6^ conidia/mL by sterile DW. The experiment was set up as flower inoculation by spraying: (1) 50 mL DW (control group), (2) 50 mL *N. clavispora* conidial suspension, (3) 50 mL *Trichoderma* strain conidial suspension, then incubated at ambient temperature for 24 h and challenged similarly with *N. clavispora* conidial suspension, and (4) 50 mL *Trichoderma* conidial suspension. Five flowers were inoculated for each treatment and the experiment was repeated twice. The tested flowers were incubated in the moist chamber (approximately 85% relative humidity) at ambient temperature, and disease development was observed after 10 days postinoculation. Disease scores were determined by the method previously described by Sunpapao et al. [9], with some modifications, based on assessing the external symptoms of the flowers (0 = no symptom, 1 = small brown symptom < 1 cm, 2 = brown symptom > 1 cm, 3 = brown symptom that covers > 25% of flower, 4 = brown symptom that covers more than 50% of flower). Disease scores were converted to a disease severity index (DSI) using the following formula:$$\mathrm{DSI}\;\left(\%\right)=\frac{\sum (\mathrm{Scale}\times \mathrm{Amount}\;\mathrm{of}\;\mathrm{plants})}{\mathrm{Maximum}\;\mathrm{level}\times \mathrm{Total}\;\mathrm{number}\;\mathrm{of}\;\mathrm{plants}}\times 100$$

### 2.8. Statistical Analysis

Significant differences among fungal growth, mycoparasite test, and disease severity were subjected to one-way analysis of variance (ANOVA). Tukey’s test and Student’s *t* test were used to determine statistically significant differences.

## 3. Results

### 3.1. Primary Screening of Trichoderma spp., against Neopestalotiopsis clavispora

Dual culture assays showed that the ten isolates of *Trichoderma* spp. grew faster in tested PDA plates compared to *N. clavispora* and covered most of the petri dish after 5 days of incubation. *Trichoderma* spp. inhibited mycelial growth of *N. clavispora*, with inhibition ranging from 59.61% to 78.20%. *Trichoderma* sp. T1-02 presented the highest inhibition against *N. clavispora*, 78.20%, significantly higher than those of other isolates (Figure 1A). At 7 days after incubation at ambient temperature, *Trichoderma* sp. T1-02 effectively covered the whole colony of *N. clavispora* (Figure 1B).

### 3.2. Identification of Trichoderma Strain T1-02

*Trichoderma* sp. T1-02 conidia germinated on PDA within 24 h and reached 9 cm in three days at ambient temperature (28 ± 2 °C), revealing a growth rate of 3 cm per day. Fungal colonies of *Trichoderma* sp. T1-02 displayed white, pale green, and turned to dark green after 5 days (Figure 2A,B). Conidiophores branched, terminating a whorl of two to three divergent phialides. Phialides were 6.22–16.38 μm long, 2.33–9.96 μm wide ($\bar{x}$ = 10.79 × 4.02 μm, *n* = 30) and apulliform. Conidia were 3.39–5.16 μm long, 3.08–4.40 μm wide ($\bar{x}$ = 4.31 × 3.78 μm, *n* = 30), and green to dark-green globose to ovule (Figure 2E).

For molecular identification, portions of ITS, *rpb2* and *tef1-α* regions gave DNA fragments about 500, 1200, and 1200 bases long, respectively. The PCR products were then deposited in the GenBank database and assigned accession numbers OR578379, OR587842, and OR587841 for ITS, *rpb2* and *tef1-α* regions, respectively, and they were compared with known sequences of *Trichoderma* spp. available at GenBank. A BLASTn search in GenBank indicated that *Trichoderma* sp. T1-02 was grouped within *Trichoderma virens* species with 100% identity of the all sequences (ITS, *rpb2* and *tef1-α* regions). The combined ITS, *rpb2* and *tef1-α* sequences dataset consisted of 44 taxa. The phylogram assigned the fungal isolate T1-02 used in this study to the same clade of *Trichoderma virens* containing species of ex-type (DAOM167652), with high BS (99%) and PP (1.0) supports. Therefore, the isolate T1-02 was identified as *T. virens* based on molecular properties (Figure 3).

### 3.3. Production of Volatile Antifungal Compounds

The sealed plate method showed that *T. virens* T1-02 inhibited the growth of *N. clavispora*, with inhibition of 51.28% in PDA plates (Figure 4A), and suppressed growth of *N. clavispora* in the tested PDA plate (Figure 4B). This result revealed that VOCs emitted by *T. virens* T1-02 were able to restrict fungal growth. SPME-GC/MS showed a major volatile emitted by *T. virens* T1-02 at 15.22 min (Figure 5). This compound was tentatively identified as a sesquiterpene: germacrene D (IUPAC name: (1*E*,6*E*,8*S*)-1-methyl-5-methylidene-8-propan-2-ylcyclodeca-1,6-diene). The germacrene D contained C 15 atoms (C_15_) with a molecular weight of 204 and molecular formula of C_15_H_24_.

### 3.4. Mycoparasitic Properties of Trichoderma T1-02

Mycoparasitic ability of *T. virens* T1-02 was assessed through the pre-colonized plate method. After incubation for 5 days, hyphae of T1-02 attaching to *N. clavispora* hyphae were observed by compound microscopy (Figure 6B). SEM showed *T. virens* T1-02 conidia attached on *N. clavispora* hyphae and caused hyphae wilt after 10 days of incubation (Figure 6C). After incubation for 3 weeks, *T. virens* T1-02 was aggressively mycoparasitic to *N. clavispora*, with colonization of 100% (Figure 6A). The pathogen could not be recovered from this point of colonization on PDA.

### 3.5. Trichoderma virens T1-02 Suppressed Symptom Development

An in vivo test was conducted to test the effect of *Trichoderma* on the reduction of flower blight on flamingo flowers. Application of the *T. virens* T1-02 conidia suspension reduced symptom development and disease severity index of flower blight on flamingo flowers (Figure 7C). After 10 days of incubation, brown lesions in control (inoculation with *N. clavispora* alone) showed the most severe symptoms compared to other treatments (Figure 7B). Disease severity index (DSI) of flamingo flowers inoculated with DW, *N. clavispora* alone, *N. clavispora* challenged with *T. virens* T1-02 and *T. virens* T1-02 alone was 0%, 85.52%, 12.50%, and 0%, respectively.

## 4. Discussion

In the current study, the newly isolated rhizosphere fungus displayed antagonistic ability against *N. clavispora* in dual culture assays. Based on morphological study and molecular properties of multiple DNA sequences of ITS, *tef1-α* and *rpb2* this rhizosphere fungus was identified as *T. virens* strain T1-02. Generally, the plant pathogens were suppressed by the rhizosphere fungi through multiple mechanisms, including antibiosis, competition, mycoparasitism, and induction of defense responses in plants [9,10,15]. Meanwhile, the possible biocontrol mechanisms of *T. virens* T1-02 against *N. clavispora* were assessed both in vitro and in vivo. Interestingly, our consistent results from the two assessments indicate that *T. virens* T1-02 has the potential to be used as a BCA in inhibiting *N. clavispora*. The possible mechanisms of *T. virens* T1-02 include competition for nutrients and space, production of VOCs, and direct parasitism on pathogen hyphae, leading to reduction in disease severity in flamingo flowers.

Based on the above results, T1-02 effectively suppressed fungal growth of *N. clavispora* on dual culture assay. This may be due to competition and/or antifungal metabolites released from *T. virens* T1-02. The results from our study showed that *T. virens* T1-02 effectively inhibited the fungal growth of *N. clavispora* in dual culture assay plates due to a faster growth rate and covering the pathogen colony. The ability to compete for nutrients and space is one of the most effective mechanisms involved in fungicidal ability against pathogens. Due to the fast-growing ability of *Trichoderma* spp. [44], they were able to compete for nutrients and space in vitro, as observed by dual culture assay [28]. This method is widely used to primarily screen antagonistic activities of several pathogens [9,14]. This suggests that *T. virens* T1-02 exhibited competition mechanism that involved antagonistic ability by restricting the fungal growth of *N. clavispora*, the flower blight pathogen of the flamingo flower.

In the current study, the sealed plate method showed that VOCs emitted by *T. virens* T1-02 effectively inhibited the fungal growth of *N. clavispora*. Similar results were observed in several *Trichoderma* species that could produce and release VOCs responsible for restricting the growth of fungal pathogens [45,46]. For instance, volatiles emitted by *T. asperelloides* PSU-P1 mediated antifungal ability, induced defense response, and promoted plant growth in *Arabidopsis thaliana* [11]. Furthermore, volatiles emitted by *T. asperellum* T76-14 contained 2-phynyl ethanol (2-PE), associated with reduction of postharvest fruit rot of muskmelon [13]. The GC/MS profile also suggested that VOCs of *T. virens* T1-02 contained germacrene D. This compound is classified into sesquiterpene, which is commonly found in plant essential oil with antimicrobial activities [47,48]. However, this compound was detected in a different strain of *T. virens* by Nieto-Jacobo et al. [49]. Our result is in agreement with Nieto-Jacobo et al. [49] that different strains of *T. virens* can produce and emit germacrene D, and may contribute to antagonistic activity against *N. clavispora*.

The pre-colonized plate method was used in this study to reveal this mycoparasitism, and *T. virens* T1-02 aggressively colonized 100% of *N. clavispora* in tested plates. This method showed that hyphae and conidia of *T. virens* T1-02 had strong inhibitory and parasitic effects on hyphae of *N. clavispora*. This ability has been found in different species of *Trichoderma* by Bailey et al. [10]. They are aggressively mycoparasitic against *Moniliophthora roreri*, the fungal pathogen of pod rot of *Theobroma cacao*. A recent publication by Runagwong et al. [14] also revealed that *T. asperelloides* PSU-P1 acts as a mycoparasite against *Stagonosporopsis cucurbitacearum*, the fungal pathogen of gummy stem blight in muskmelon. The finding from the present study suggested that the *T. virens* T1-02 not only competed for nutrients and spaces but also parasitized and colonized fungal hyphae of *N. clavispora*.

This study aimed to reduce the excess use of synthetic fungicides in agricultural processes and to find environmentally friendly ways to combat *N. clavispora* in flamingo flowers. The results from in vivo testing revealed that the application of rhizosphere fungus *T. virens* T1-02 conidia suspension reduced the disease severity of flower blight in flamingo flowers caused by *N. clavispora* when compared to the control without rhizosphere fungus inoculation. As observed by Wonglom et al. [50], application of *Trichoderma* sp. T76-12/2 conidia suspension reduced disease severity of *Sclerotium* fruit rot of snake fruit and stem rot of lettuce. Furthermore, the development of an emulsion formulation of *T. asperelloides* PSU-P1 has been noted as an effective formulation in suppressing stem canker of dragon fruit [51]. Furthermore, the application of *T. virens* 6PS-2 spore suspensions changed the community of microorganisms through colonization in rhizosphere soil and reduced a pathogen of apples [52]. Our current study provides evidence on the possibilities of applying *T. virens* T1-02 as BCA for managing *Neopestalotiopsis* diseases including flower blight of flamingo flower, which would be an eco-friendly management strategy of *N. clavispora* and contribute to the healthy maintenance of plants.

## 5. Conclusions

The rhizosphere soil fungus was primarily screened for its fungicidal activity against *N. clavispora* and identified based on both morphology and molecular data as *T. virens* T1-02. The results revealed that this strain of *T. virens* has the potential to be used as a biological control against *N. clavispora* with multiple mechanisms, including antibiosis via the production of volatile antifungal compound germacrene D, competition and direct parasitism. An in vivo test suggested the potential of *T. virens* T1-02 to reduce the disease severity of flower blight on flamingo flowers. However, the bioformulation of this strain has yet to be investigated. To apply *T. virens* T1-02 as BCA on a large scale for commercial purposes, further experiments need to be verified in the near future.

## Figures and Tables

**Figure 1 jof-10-00066-f001:**
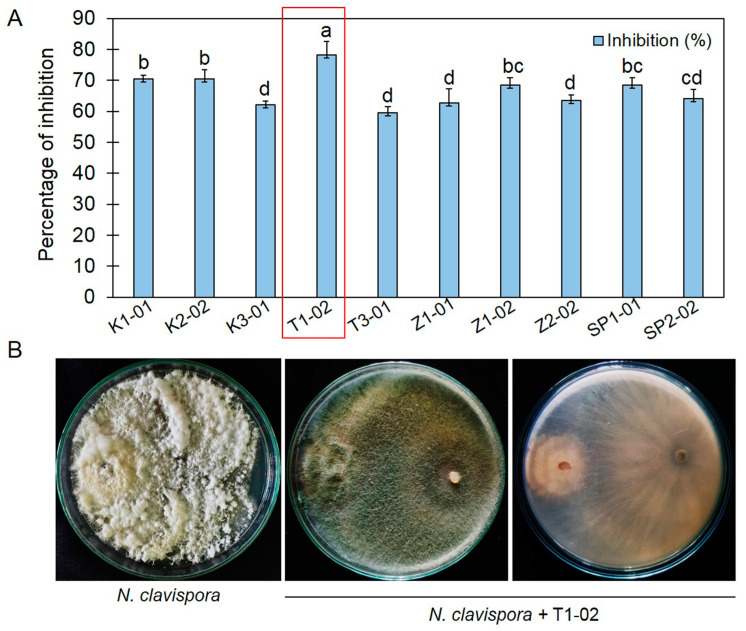
Dual culture assays of *Trichoderma* spp. Percentage inhibition of *Trichoderma* spp. against *Neopestalotiopsis clavispora* (**A**) and dual culture assay plates containing control (**left**), tested plates with *N. clavispora* and *Trichoderma* sp. T1-02 in top (**middle**) and bottom (**right**) view (**B**). Error bars indicate standard deviation (SD) of percentage of inhibition (%), whereas letters indicate significant difference among treatments according to Tukey’s test (*p* < 0.05).

**Figure 2 jof-10-00066-f002:**
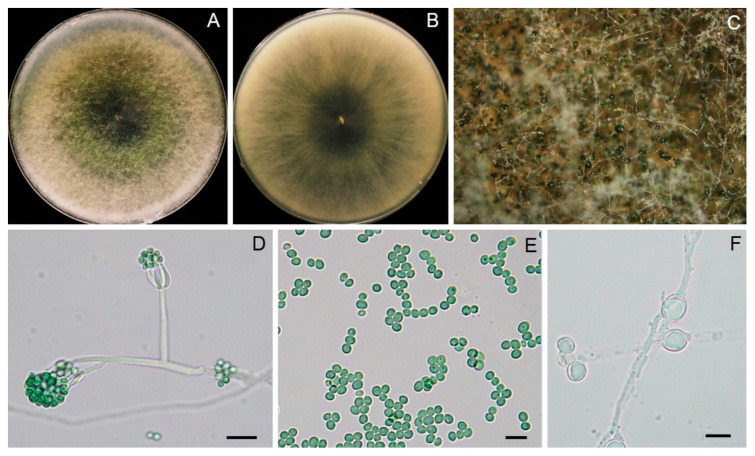
Morphological characteristics of T1-02, colony on PDA in top (**A**) and bottom view (**B**); pustule conidiation and mass of conidia developed on tip of conidiophores observed by stereomicroscope (**C**); details of conidiophore, phialides and conidia (**D**), conidia (**E**) and chlamydospores (**F**).

**Figure 3 jof-10-00066-f003:**
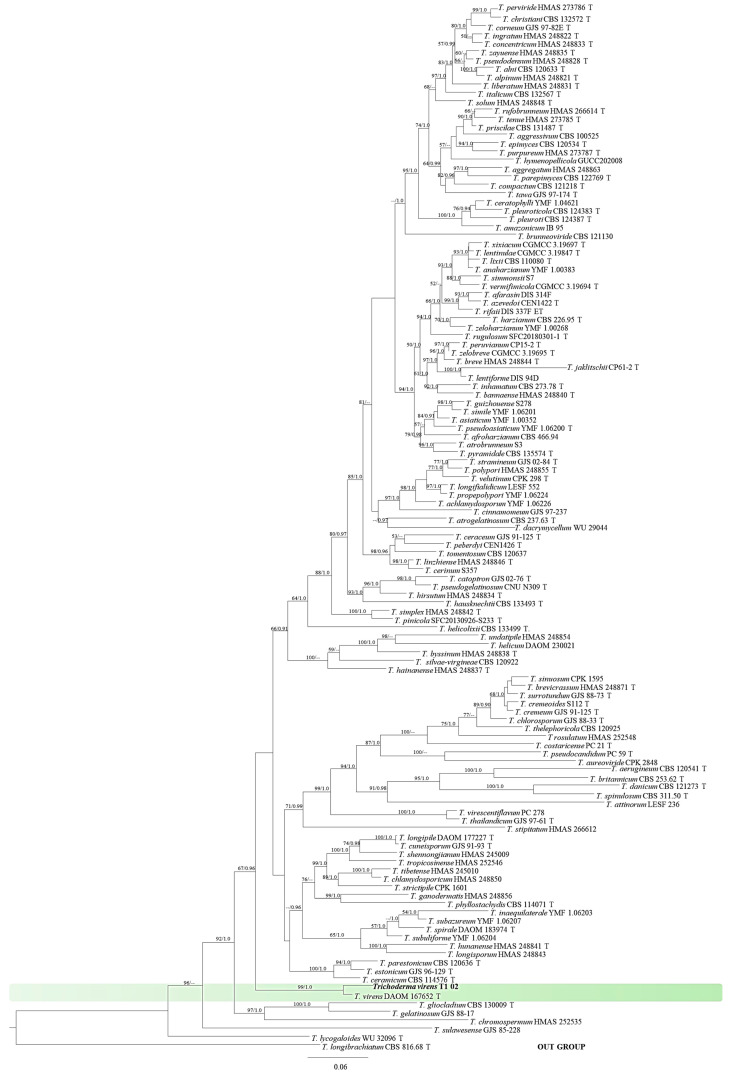
Phylogram derived from maximum-likelihood (ML) method of combined ITS, *tef1-α* and *rpb2* sequences of 122 taxa with *T. longibrachiatum* used as the outgroup. Maximum-likelihood bootstrap support (BS) above 50% and Bayesian posterior probability (PP) above 0.9 are shown at nodes. Scale bar represents 6 substitutions per nucleotide position. The new isolate obtained in this study is in bold. “T” represents ex-type strains.

**Figure 4 jof-10-00066-f004:**
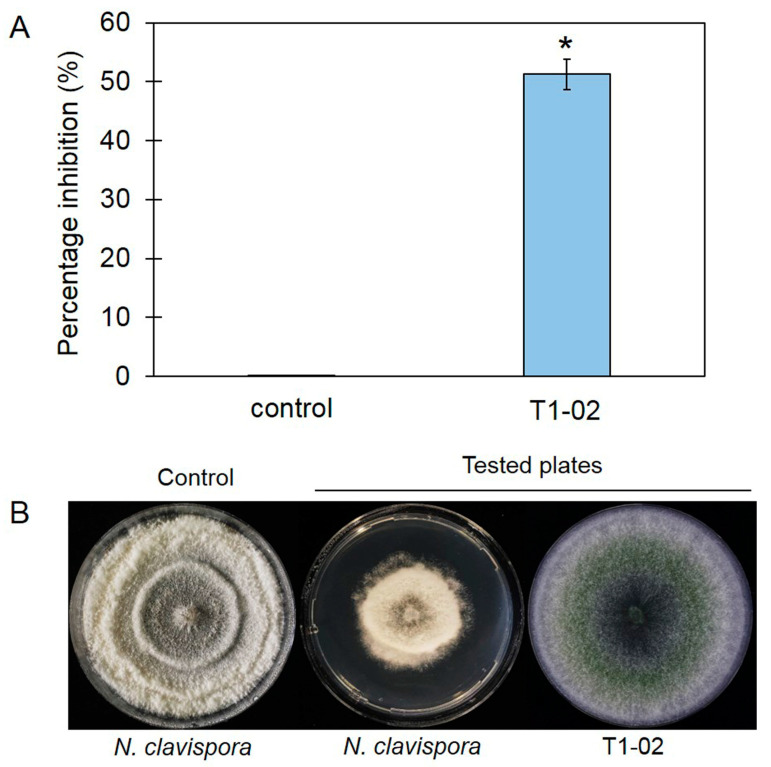
Volatile antifungal bioassay, percentage inhibition of *Trichoderma virens* T1-02 against *Neopestalotiopsis clavispora* (**A**), and sealed plate method on tested plate containing control and tested plates (**B**). Error bars indicate standard deviation (SD) of percentage inhibition (%), whereas asterisk (*) indicates significant difference between treatments according to Student’s *t* test (*p* < 0.05).

**Figure 5 jof-10-00066-f005:**
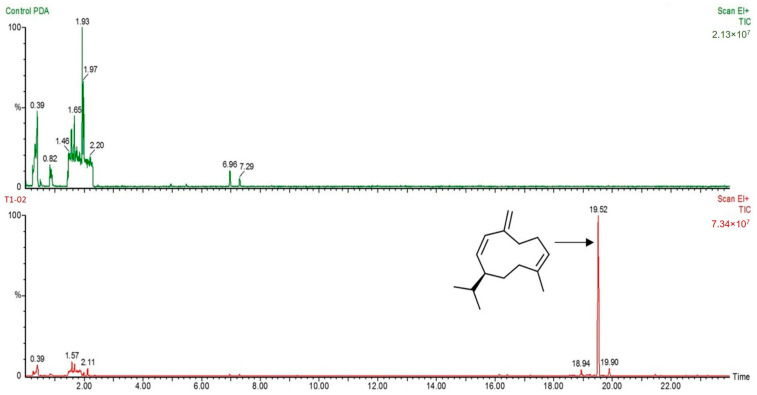
GC/MS profiling of PDA (control; **up**) and volatiles emitted by *Trichoderma virens* T1-02 (**down**), major peak at 19.52 min tentatively identified as germacrene D and its structure.

**Figure 6 jof-10-00066-f006:**
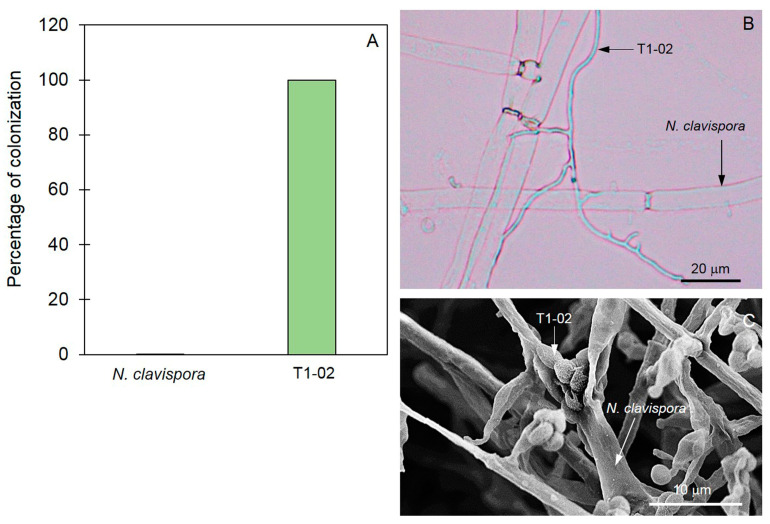
Percentage colonization of *Trichoderma virens* T1-02 against *Neopestalotiopsis clavispora* by pre-colonized plate method (**A**), hyphae of T1-02 attached on *N. clavispora* hyphae observed by compound microscope (**B**), conidia of T1-02 attached on *N. clavispora* hyphae observed by SEM (**C**).

**Figure 7 jof-10-00066-f007:**
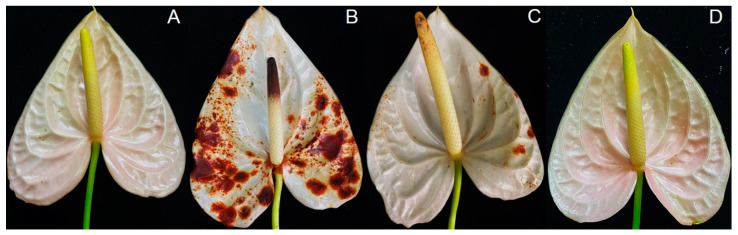
Flower blight disease development of flamingo flower inoculated with DW (**A**), *Neopestalotiopsis clavispora* alone (**B**), *N. clavispora* challenged with T1-02 (**C**) and T1-02 alone (**D**).

## Data Availability

The DNA sequence data obtained from this study were deposited in GenBank under accession numbers ITS (OR578379), *rpb2* (OR587842) and *tef1*-α (OR587841).

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
