# Peer review of "Biological Control Activities of Rhizosphere Fungus *Trichoderma virens* T1-02 in Suppressing Flower Blight of Flamingo Flower (*Anthurium andraeanum* Lind.)"

_jof, 2024, doi:10.3390/jof10010066_

Round 1

Reviewer 1 Report

Comments and Suggestions for Authors

Dusit Athinuwat et al. isolated and screened a biocontrol strain T1-02 with significant inhibitory effects on Neopestalotiopsis clavispora. And identified it as Trichoderma viride through morphological and molecular phylogenetic analysis. The author demonstrated through experiments that strain T1-02 can produce volatile antibacterial substances and obtained an antifungal compound (Germacrene) by GC-MS analysis. In addition, strain T1-02 can also directly parasitize on pathogenic fungal hyphae. Furthermore, application of T. virens T1-02 spore suspension reduced disease severity index of flower blight in flamingo flower. This study provides a potential biocontrol strain for the prevention and control of flower blight in flamingo flower. I would like to address some points below:

1.        Line 284-286: SEM micrograph showed T. virens T1-02 conidia attached on N. clavispora hyphae and caused hyphae wilt after 10 days of 285 incubation (Fig. 4). But the results described in this section correspond to Figure 6C.

2.        Suggest adding arrows on Figure 6C to indicate the phenomenon to be explained.

3.        Line 193-194: The percentage of colonization was calculated. Need to add a method or formula for statistical planting percentage

4.        Line 242-244: Annotations need to be added for Figure 2C.

5.        Figures 1, 2, 4, 6, and 7 in the text are all composed of multiple parts. It is recommended to distinguish them when describing different parts of the same image. For example, in section 3.4, the results of Figure 6A, Figure 6B, and Figure 6C were described separately, but they were not distinguished when annotated. Suggest replacing Figure 6 in parentheses at line 284 with Figure 6B, replacing Figure 6 in parentheses at line 286 with Figure 6C, and replacing Figure 6 in parentheses at line 287 with Figure 6A.

Comments on the Quality of English Language

Minor editing of English language required

Author Response

Reviewer 1

Dusit Athinuwat et al. isolated and screened a biocontrol strain T1-02 with significant inhibitory effects on Neopestalotiopsis clavispora. And identified it as Trichoderma viride through morphological and molecular phylogenetic analysis. The author demonstrated through experiments that strain T1-02 can produce volatile antibacterial substances and obtained an antifungal compound (Germacrene) by GC-MS analysis. In addition, strain T1-02 can also directly parasitize on pathogenic fungal hyphae. Furthermore, application of T. virens T1-02 spore suspension reduced disease severity index of flower blight in flamingo flower. This study provides a potential biocontrol strain for the prevention and control of flower blight in flamingo flower. I would like to address some points below:

Answer: Thank you for comments this manuscript and suggested valuable comments to improve this manuscript.

  1. Line 284-286: SEM micrograph showed T. virens T1-02 conidia attached on N. clavispora hyphae and caused hyphae wilt after 10 days of 285 incubation (Fig. 4). But the results described in this section correspond to Figure 6C.

Answer: We have revised as Fig. 6.

  1. Suggest adding arrows on Figure 6C to indicate the phenomenon to be explained.

Answer: We have added arrows on Fig. 6C as suggestion.

  1. Line 193-194: The percentage of colonization was calculated. Need to add a method or formula for statistical planting percentage

Answer: For this method, percentage of colonization was calculated by counting growth of colony Trichoderma in comparison with colony grpwth of N. clavispora in 20% PDA plates.

  1. Line 242-244: Annotations need to be added for Figure 2C.

Answer: We have added “pustule conidiation and mass of conidia developed on tip of conidiophores observed by stereomicroscope”.

  1. Figures 1, 2, 4, 6, and 7 in the text are all composed of multiple parts. It is recommended to distinguish them when describing different parts of the same image. For example, in section 3.4, the results of Figure 6A, Figure 6B, and Figure 6C were described separately, but they were not distinguished when annotated. Suggest replacing Figure 6 in parentheses at line 284 with Figure 6B, replacing Figure 6 in parentheses at line 286 with Figure 6C, and replacing Figure 6 in parentheses at line 287 with Figure 6A.

Answer: We have revised according to reviewer comments in description of Fig. 1, 2, 4, 6 and 7 in the text.

Reviewer 2 Report

Comments and Suggestions for Authors

Dear authors

I reviewed the manuscript ID jof-2806473 -entitled “Biological Control Activities of a Rhizosphere Fungus Trichoderma virens T1-02 in Suppressing Flower Blight of Flamingo
Flower (Anthurium andraeanum) " for its publication on Journal of Fungi.

In the first place, I apologize if my English is not very good because is not my mother tongue. Please excuse any errors on my part.

The manuscript is focused on the search on the harmless alternative to manage an emerging disease on flower blight cause by Neopestalotiopsis clavispora. Ten strains of Trichoderma have been screened and one of them has been selected.

The authors performed in vitro and in vivo assays with the aim to evaluate this strain and its mechanisms of action to biocontrol the pathogen. 

I consider that the manuscript could contribute to the development of biopesticides, which is an attractive option for management of plant diseases.

However, in my opinion there are in the manuscript some issues that are not clear and others I consider could be clarified to make it more understandable. There are issues in the methodology that could be clarified.

I´m listed below corrections, questions or suggestions that I think should be made in the text to improve or clarify the manuscript for publication.

I hope that my comments and suggestions will contribute to improving the manuscript.

Some others suggestions have been made in the manuscript.

Sincerely,

SL

Abstract: Lines 21-22. This research aimed to screen the potential of rhizosphere fungi, Trichoderma spp. with diverse antifungal abilities to control N. clavispora and to reduce flower blight in flamingo flowers.

Line 59: Flamingo flower (Anthurium andraeanum Lind.)

Lines 84-86: …“Therefore, the aim of this research is to screen rhizosphere Trichoderma species against N. clavispora, the pathogen of flower blight of flamingo flowers”…..

I consider that the objectives should be amplified and the authors should be mention that they want study the mechanisms of action involved by the biocontrol agent.

Line 88: ….2.1. Sources of Trichoderma and Pathogen”…. In my opinion the subtitle could be changed by “Source of Strains of Trichoderma and the Pathogen”

Lines 89-94: The strains of Trichoderma were obtained from the Culture Collection of Pest Management Department or they were isolated from rhizosphere soil if Para rubber three. It is not clear, if the strains were previously isolated from rhizosphere and then deposited on the Culture Collection.

Line 109: I consider that the authors can mention the criteria for selecting the strains of Trichoderma.

Line 111: Macroscopic and microscopic features of Trichoderma strain were observed by stereomicroscope..

Lines 151-153: The authors proposed the volatile antifungal bioassay in order to study the ability of the Trichoderma strains to produce volatile compound to inhibit the growth of N. clavispora. This should be clarified in the initial sentence because as it is written the authors assume that this strain produces volatile compounds

Lines 193-194: The percentage of colonization of each fungus was calculated.

Line 201: I suggest change “the effective Trichoderma isolate” by “the selected Trichoderma isolate”.

Line 202: I suggest to clarified “ambient temperature”. How were fungi cultivated? On Petri plates on PDA??

Line 203: The authors could mention that suspensions of the pathogen and of the Trichoderma strains were prepared and that suspension were adjusted.

Suspensions were adjusted using the Neubauer chamber?

Line 204: How was performed the control?

Line 205: …..3) flower inoculation with Trichoderma strain  by spraying (50 mL) ….

Line 205: Please clarified: 3) flower inoculation with Trichoderma by spraying (50 mL) and in-cubated at ambient temperature for 24 h and challenged with N. clavispora in the same 206
method,…….

This sentence it is not clear to me..This sentence means that flowers were treated preliminary with the strain of Trichoderma and then these were treated with the pathogen???

Were made humid chamber after inoculations?

Line 222: Primary Screening of Trichoderma spp. Against Neopestalotiopsis clavispora

Line 233: I suggest to change the subtitle “Trichoderma T1-02 Suppressed Symptom Development” by “Identification of Trichoderma strain T1-02”

Figure 6. It would be clearer if arrows were included marking the attaching

Author Response

Reviewer 2

Dear authors

I reviewed the manuscript ID jof-2806473 -entitled “Biological Control Activities of a Rhizosphere Fungus Trichoderma virens T1-02 in Suppressing Flower Blight of Flamingo
Flower (Anthurium andraeanum) " for its publication on Journal of Fungi.

In the first place, I apologize if my English is not very good because is not my mother tongue. Please excuse any errors on my part.

The manuscript is focused on the search on the harmless alternative to manage an emerging disease on flower blight cause by Neopestalotiopsis clavispora. Ten strains of Trichoderma have been screened and one of them has been selected.

The authors performed in vitro and in vivo assays with the aim to evaluate this strain and its mechanisms of action to biocontrol the pathogen. 

I consider that the manuscript could contribute to the development of biopesticides, which is an attractive option for management of plant diseases.

However, in my opinion there are in the manuscript some issues that are not clear and others I consider could be clarified to make it more understandable. There are issues in the methodology that could be clarified.

I´m listed below corrections, questions or suggestions that I think should be made in the text to improve or clarify the manuscript for publication.

I hope that my comments and suggestions will contribute to improving the manuscript.

Some others suggestions have been made in the manuscript.

Sincerely,

SL

Answer: Thank you for your review and gave us a valuable suggestion to improve this manuscript.

Abstract: Lines 21-22. This research aimed to screen the potential of rhizosphere fungi, Trichoderma spp. with diverse antifungal abilities to control N. clavispora and to reduce flower blight in flamingo flowers.

Answer: We have revised as suggestion.

Line 59: Flamingo flower (Anthurium andraeanum Lind.)

Answer: We have added author name as Anthurium andraeanum Lind.

Lines 84-86: …“Therefore, the aim of this research is to screen rhizosphere Trichoderma species against N. clavispora, the pathogen of flower blight of flamingo flowers”…..

I consider that the objectives should be amplified and the authors should be mention that they want study the mechanisms of action involved by the biocontrol agent.

Answer: We have revised as “Therefore, the aim of this research is to screen and to test fungicidal activities of rhizosphere Trichoderma species against N. clavispora, the pathogen of flower blight of flamingo flowers.”

Line 88: ….2.1. Sources of Trichoderma and Pathogen”…. In my opinion the subtitle could be changed by “Source of Strains of Trichoderma and the Pathogen”

Answer: We have revised as Source of Strains of Trichoderma and the Pathogen

Lines 89-94: The strains of Trichoderma were obtained from the Culture Collection of Pest Management Department or they were isolated from rhizosphere soil if Para rubber three. It is not clear, if the strains were previously isolated from rhizosphere and then deposited on the Culture Collection.

Answer: Actually, all Trichoderma strains were previously isolated from rhizosphere soil, and then deposited in Culture Collection. Therefore we revised as “A total of ten Trichoderma strains namely K1-01, K2-02, K3-01, T1-01, T3-01, Z1-01, Z1-02, Z2-02, SP1-01 and SP2-02 were previously isolated from rhizosphere soil of Para rubber tree (Hevea brasiliensis) at plant protected area, Songkhla province, southern Thailand by soil dilution pour plate in a previous study [6] and deposited in Culture Collection of Pest Management Department, Faculty of Natural Resources, Prince of Songkla University, Thailand.”

Line 109: I consider that the authors can mention the criteria for selecting the strains of Trichoderma.

Answer: We have added criteria for this identification as “Selected Trichoderma isolate with the highest percentage inhibition against N. clavispora was cultured on PDA…”

Line 111: Macroscopic and microscopic features of Trichoderma strain were observed by stereomicroscope.

Answer: We have revised as suggestion.

Lines 151-153: The authors proposed the volatile antifungal bioassay in order to study the ability of the Trichoderma strains to produce volatile compound to inhibit the growth of N. clavispora. This should be clarified in the initial sentence because as it is written the authors assume that this strain produces volatile compounds

Answer: We have added “Several Trichoderma strains are able to produce and release volatile organic compounds which responsible for suppressing fungal growth.” for the first paragraph for this sentence.

Lines 193-194: The percentage of colonization of each fungus was calculated.

Answer: We have revised as suggestion.

Line 201: I suggest change “the effective Trichoderma isolate” by “the selected Trichoderma isolate”.

Answer: We have revised as suggestion.

Line 202: I suggest to clarified “ambient temperature”. How were fungi cultivated? On Petri plates on PDA??

Answer: We have revised as “The selected Trichoderma isolate and N. clavispora were cultured on PDA for 7 days and incubated in ambient temperature (28±2°C) for sporulation.”

Line 203: The authors could mention that suspensions of the pathogen and of the Trichoderma strains were prepared and that suspension were adjusted.

Suspensions were adjusted using the Neubauer chamber?

Answer: We have revised as “Conidia of both Trichoderma and N. clavispora were harvested. Conidial suspension of each fungus was adjusted concentration as 106 conidia/mL by sterile DW.”

Line 204: How was performed the control?

Answer: We have revised as “flower inoculation with DW by spraying (50 mL) was served as control group”

Line 205: …..3) flower inoculation with Trichoderma strain  by spraying (50 mL) ….

Answer: We have revised as suggestion.

Line 205: Please clarified: 3) flower inoculation with Trichoderma by spraying (50 mL) and in-cubated at ambient temperature for 24 h and challenged with N. clavispora in the same 206
method,…….

This sentence it is not clear to me. This sentence means that flowers were treated preliminary with the strain of Trichoderma and then these were treated with the pathogen???

Answer: In order to test Trichoderma strain could protect flower from pathogen, this treatment Trichoderma was preliminary treated on flower, followed by inoculation of pathogen.

Were made humid chamber after inoculations?

Answer: We have revised as “The tested flowers were incubated in moist chamber (approximately 85% relative humidity) at ambient temperature.”

Line 222: Primary Screening of Trichoderma spp. Against Neopestalotiopsis clavispora

Answer: We have revised as suggestion.

Line 233: I suggest to change the subtitle “Trichoderma T1-02 Suppressed Symptom Development” by “Identification of Trichoderma strain T1-02”

Answer: We have revised as suggestion.

Figure 6. It would be clearer if arrows were included marking the attaching

Answer: We have included arrows and marking in Fig. 6C.

Reviewer 3 Report

Comments and Suggestions for Authors

The manuscript by Athinuwat et al. nicely presented multiple experiments to prove the beneficial effects of T. virens T1-02. The manuscript is well fitted with the Journal of Fungi. However, I think the authors should correct or clarify a few points that will potentially improve the manuscript.

Line 55- several tropical diseases such as -give some examples.

Line 84- Potential typo in "plant pathology filed". It must be "field".

Line 93- Do you have more info regarding the pathogen strains such as strain name and origin.

Line 156- a plate "g"??? Not sure what the authors meant.

Line 189- 7 7- is it a typo?

Line 287- How did you measure the rate of colonization? 

In vivo test section is not really clear what the authors did. Did you apply Trichoderma directly to flower or soil? Did you eventually detach the flowers? 

Citation 41-42 is messed up. Please check the citation and correct the issue.

Figure 1 B- not sure what is the second and the third pics. 

Figure 3- low resolution on the highlighted part of the tree

Comments on the Quality of English Language

Minor issues that can easily be fixed.

Author Response

Reviewer 3

The manuscript by Athinuwat et al. nicely presented multiple experiments to prove the beneficial effects of T. virens T1-02. The manuscript is well fitted with the Journal of Fungi. However, I think the authors should correct or clarify a few points that will potentially improve the manuscript.

Answer: Thank you for your comments and suggested valuable comments to improve this manuscript. We have revised this manuscript according to reviewer’ comments in blue text throughout this manuscript.

Line 55- several tropical diseases such as -give some examples.

Answer: We have added “such as stem canker of dragon fruit [15], gummy stem blight of muskmelon [16] and Cercospora leaf spot of sugar beet [18]

Line 84- Potential typo in "plant pathology filed". It must be "field".

Answer: We have revised as suggestion.

Line 93- Do you have more info regarding the pathogen strains such as strain name and origin.

Answer: We have revised as “The relevant Trichoderma and flower blight pathogen N. clavispora strain PSU-AAN2, which isolated from flamingo flower in Thailand…”

Line 156- a plate "g"??? Not sure what the authors meant.

Answer: We have removed “g” from this sentence.

Line 189- 7 7- is it a typo?

Answer: We have revised as “7-day old colony…”

Line 287- How did you measure the rate of colonization? 

Answer: Percentage of colonization was measured by the growth of Trichoderma or N. clavispora on 20% PDA plates cut from pre-colonized tested plates.

In vivo test section is not really clear what the authors did. Did you apply Trichoderma directly to flower or soil? Did you eventually detach the flowers? 

Answer: We have directly applied Trichoderma spore suspension onto flower by detach the flowers.

Citation 41-42 is messed up. Please check the citation and correct the issue.

Answer: We have revised citation as suggestion.

Figure 1 B- not sure what is the second and the third pics. 

Answer: Fig 1 B, culture assay  plates containing control (left), tested plates in top (middle) and bottom (right) view (B).

Figure 3- low resolution on the highlighted part of the tree

Answer: We have replaced with new figure with high resolution.

Comments on the Quality of English Language

Minor issues that can easily be fixed.

Answer: We have carefully rechecked quality of English throughout this manuscript.